# Maternal Chronic Ultrasound Stress Provokes Immune Activation and Behavioral Deficits in the Offspring: A Mouse Model of Neurodevelopmental Pathology

**DOI:** 10.3390/ijms241411712

**Published:** 2023-07-20

**Authors:** Dmitrii Pavlov, Anna Gorlova, Abrar Haque, Carlos Cavalcante, Evgeniy Svirin, Alisa Burova, Elizaveta Grigorieva, Elizaveta Sheveleva, Dmitry Malin, Sofia Efimochkina, Andrey Proshin, Aleksei Umriukhin, Sergey Morozov, Tatyana Strekalova

**Affiliations:** 1Hotchkiss Brain Institute, University of Calgary, Calgary, AB T2N 1N4, Canada; abrar.haque@ucalgary.ca; 2Laboratory of Cognitive Dysfunctions, Institute of General Pathology and Pathophysiology, Russian Academy of Medical Sciences, 125315 Moscow, Russia; gorlova_a_v@staff.sechenov.ru (A.G.); svirin_e_p@staff.sechenov.ru (E.S.); burova.ae@phystech.edu (A.B.); grigorieva.es@phystech.edu (E.G.); shevelevalisa02@gmail.com (E.S.); niiopp@mail.ru (S.M.); tatslova@gmail.ru (T.S.); 3Department of Human Health and Science, MacEwan University, Edmonton, AB T5J 4S2, Canada; dearaujocavalcantejuniorc@mymacewan.ca; 4Laboratory of Psychiatric Neurobiology, Department of Normal Physiology, Institute of Molecular Medicine, Sechenov First Moscow State Medical University, 119991 Moscow, Russia; dmalin@medicine.wisc.edu (D.M.); efimochkina.sofi@gmail.com (S.E.); alum1@yandex.ru (A.U.); 5P.K. Anokhin Research Institute of Normal Physiology, 125315 Moscow, Russia; proshin_at@mail.ru

**Keywords:** ultrasound radiation, systemic inflammation, pro-inflammatory cytokines, depression, memory, offspring, mice

## Abstract

Neurodevelopmental disorders stemming from maternal immune activation can significantly affect a child’s life. A major limitation in pre-clinical studies is the scarcity of valid animal models that accurately mimic these challenges. Among the available models, administration of lipopolysaccharide (LPS) to pregnant females is a widely used paradigm. Previous studies have reported that a model of ‘emotional stress’, involving chronic exposure of rodents to ultrasonic frequencies, induces neuroinflammation, aberrant neuroplasticity, and behavioral deficits. In this study, we explored whether this model is a suitable paradigm for maternal stress and promotes neurodevelopmental abnormalities in the offspring of stressed females. Pregnant dams were exposed to ultrasound stress for 21 days. A separate group was injected with LPS on embryonic days E11.5 and E12.5 to mimic prenatal infection. The behavior of the dams and their female offspring was assessed using the sucrose test, open field test, and elevated plus maze. Additionally, the three-chamber sociability test and Barnes maze were used in the offspring groups. ELISA and qPCR were used to examine pro-inflammatory changes in the blood and hippocampus of adult females. Ultrasound-exposed adult females developed a depressive-like syndrome, hippocampal overexpression of GSK-3β, IL-1β, and IL-6 and increased serum concentrations of IL-1β, IL-6, IL-17, RANTES, and TNFα. The female offspring also displayed depressive-like behavior, as well as cognitive deficits. These abnormalities were comparable to the behavioral changes induced by LPS. The ultrasound stress model can be a promising animal paradigm of neurodevelopmental pathology associated with prenatal ‘emotional stress’.

## 1. Introduction

Systemic chronic inflammation contributes to the development of many pathological states including cancers, neurodegenerative processes, autoimmune diseases, and mood disorders [1]. Parental chronic inflammation may be transmitted to the offspring via a process known as DIS (DNA inflammatory signature) through epigenetic alterations that result in the increased disease risk in the next generation [2]. Especially detrimental to a fetus can be maternal inflammation during pregnancy, as there is a critical window in development vulnerable to immune challenges [2]. Historically, maternal inflammation and immune activation have been centered around infections that pose a risk of neurodevelopmental abnormalities in offspring [1]. Ongoing research provides emerging evidence that maternal systemic chronic inflammation is capable of triggering robust pathologies in embryos. Animal models of parental stress, depression, obesity, asthma, and other risk factors of chronic inflammation pose risks for the offspring comparable to that of an infection [1,3]. Pro-inflammatory triggers of various nature evoke cytokine signaling cascades mediated by the placenta [4,5,6].

A large body of evidence suggests that neurodevelopment before and after birth, as well as behavior during the postnatal period, are vulnerable to cytokines and regulated by them in health and disease [7,8]. Cell-culture experiments using both human and animal samples have proved that cytokines regulate various neuroplasticity aspects such as synaptogenesis, pruning, myelination, cell migration, angiogenesis, and neurogenesis [7,9]. Knock-out animal models suggest that mice deficient in cytokines or their receptor genes are vulnerable to behavioral deficits and cognitive aberrations [10]. Importantly, maternal immune activation (MIA) during pregnancy increases the incidence of neuropsychiatric disorders in children [11,12] and is frequently utilized as a model of neurodevelopmental pathologies, including autism spectrum disorder (ASD) [7,13]. Unfortunately, many advances in our understanding of offspring behavioral abnormalities gained from animal models cannot be fully extrapolated to humans due to multiple limitations. A major confounding factor is the insufficiency of animal models to recapitulate specific human gene interactions that underlie neurodevelopmental abnormalities [9]. Another problem is that no animal model of offspring behavioral abnormalities is based on environmental insults relevant to human society. In the models of MIA there is a move to mimic viral or bacterial infection by the use of poly I:C or lipopolysaccharide (LPS) [14], but this situation is not as common in economically developed countries. At the same time, there has been a reported increase in neurodevelopmental pathologies over the last decade; for example, ASD is an emerging health threat to modern society [15].

To date, one of the most extensively utilized models of neurodevelopmental pathologies involves the use of lipopolysaccharide (LPS) during the critical window of embryonic development, which occurs at day 12.5 following the plug check [14]. Changes of molecular, structural, and behavioral levels of those pups typically show similarities with changes observed in autistic children [14,16]. For example, elevated levels of several cytokines including interferon (IFN)-γ, interleukin (IL)-1β, or IL-6 in maternal blood during pregnancy are associated with increased neurodevelopmental pathologies for their children [17]. This observation is mirrored in LPS-challenged mouse models of ASD [18,19]. Importantly, changes provoked by an immune activation are irreversible in the fetus [20] and thus persist in the form of altered behavior throughout the life of the progeny. Iconic behavioral abnormalities comprise aberrant forms of social behavior and pro-depressant symptoms [21,22]. Despite well-characterized MIA-induced neurodevelopmental changes in animal models, there is modest progress in treatment strategies of neurodevelopmental pathologies in humans. The key limitation of MIA rodent models is the artificial nature of an immune challenge that normally never happens with pregnant women in real life.

In most cultures, ‘emotional stress’ has become an omnipresent risk factor that affects human quality of life across wide geographical areas [23]. The term ‘emotional stress’ or similar definitions, i.e., ‘psychological stress’ or ‘mental stress’, are often used to label a state of stress that results from adversities in human life, such as death of relatives, divorce, humiliation or defeat, loss of social status, or deterioration of financial status [24,25]. This form of stress is considered a response to an unpredictable and mentally debilitating experience that sharply increases the risk of major depression and various psychiatric disorders. While other types of stressors are also associated with negative emotions, such as increased signs of anxiety and depressive-like behaviors after physical stress, pain, or externally induced increases in inflammatory factors or glucocorticoid levels, these stress states are not triggered by the perception and processing of adverse information and do not involve a cognitive element as a key feature. Importantly, emotional stress can provoke pro-inflammatory changes in key limbic structures [26,27] and systemically in the blood [28].

While it is not considered possible to mimic the neurobiology of ‘emotional stress’ experienced by humans in other species, we have developed a model of ‘emotional stress’ using ultrasound exposure that is more ethologically valid than many approaches [29,30,31]. In the model, mice were subjected to unpredictable exposure to ‘emotionally negative’ ultrasound signals over a 21-day period. We showed that exposure of male mice to ultrasound that unpredictably alternated in frequency between 20 and 25 kHz, which corresponded to their vocalization response to a ‘negative emotional state’, and frequencies of 25–45 kHz, which were associated with a ‘neutral’ emotional state, generated depressive-like and anxiety-like behavioral changes [32]. Although the nature of species-specific information transmitted by rodents at the ultrasonic range is not entirely clear, it has been found that mice are sensitive to sounds within certain frequency ranges. For instance, mice emit sounds in the range of 20–25 kHz in life-threatening situations, such as social defeat, pain, or maternal separation [33,34,35,36]. Therefore, exposure to artificially generated ultrasound within these frequencies may provide, in a mouse, a closer analogy to major human stressors causing depression [29,30]. The ultrasound exposure in male mice was associated with upregulated corticosterone level [30], activated hippocampal microglia, upregulated IL-1β and IL-6 locally in the hippocampus and in systemic circulation [30], and increased content of a marker of oxidative stress protein carbonyl in limbic structures [29]. Notably, ultrasound-exposed male mice developed prominent depressive-like and anxiety-like behavioral changes [29,30,31,37,38] that were counteracted by chronic dosing with the classic antidepressant fluoxetine [39]. 

Here, we sought to investigate whether the chronic exposure to the ultrasound could affect female mice similarly to males, and, if so, when applied in pregnant dams, it could induce pathological changes in the offspring. We hypothesized that anticipated immune activation in pregnant dams might result in behavioral abnormalities in the offspring, which were reminiscent of autistic-like abnormalities. Therefore, we subjected pregnant dams to the 21-day ultrasound stress test, and, using the sucrose preference test, the open field test and elevated plus maze, investigated whether this exposure could evoke the common hallmarks of a depressive-like syndrome (Figure 1A). We also studied their serum cytokine concentrations and the gene expression of GSK3β, IL-6, and IL-1β in the hippocampus. Then, we exposed pregnant dams to a 21-day ultrasound stress and investigated their female offspring behaviorally in adulthood. To compare the effects of ultrasound stress on the offspring with the effects of the LPS challenge, commonly used models of immune-driven developmental neuropathologies [18,19], we studied the effects, on the offspring, of an LPS administration to pregnant dams. Congruent batteries of the tests were employed across all behavioral studies for emotionality; additionally, in the experiments with the offspring, special learning was investigated in the Barnes maze (Figure 1B,C).

## 2. Results

### 2.1. Chronic Ultrasound Exposure Induces Behavioral Abnormalities in Adult Female Mice

In the sucrose preference test, stressed mice displayed a significant drop in sucrose preference in comparison to non-stressed animals (U = 11, *p* = 0.036, Mann–Whitney test, Figure 2A) this was a sign of decreased reward sensitivity. In the open field, ultrasound stress decreased locomotor activity (U = 9, *p* = 0.024, Figure 2B) and the time spent in the center (U = 16, *p* = 0.004, Figure 2C) in comparison with the control group. These changes in the central activity could be regarded as signs of anxiety-like behavior. In the elevated plus maze, stressed females did not differ from control animals in the number of open-arm entries (U = 8, *p* = 0.396, Figure 2D), but spent significantly less time in the open arms (U = 18, *p* = 0.015, Figure 2E), which again suggests elevated anxiety displayed by the ultrasound-exposed mice.

### 2.2. Chronic Ultrasound Exposure Induces Pro-Inflammatory Changes in Adult Female Mice

Chronic ultrasound stress exposure upregulated the content of many pronounced pro-inflammatory markers in the systemic circulation (Figure 3A), such as IL-1β (U = 0, *p* = 0.0079), IL-6 (U = 1, *p* = 0.0159), IL-17 (U = 2, *p* = 0.0317), RANTES (U = 2, *p* = 0.03), and TNFα (U = 2, *p* = 0.0397). Concentrations of IFNγ (U = 4, *p* = 0.0952), IL-1α (U = 4, *p* = 0.09), IL-12p70 (U = 5, *p* = 0.1508), IL-3 (U = 11, *p* = 0.8413), IL-4 (U = 12, *p* = 0.9), IL-5 (U = 9, *p* = 0.5476), MCP-1 (U = 10, *p* = 0.6905), and MIP-1α (U = 12, *p* = 0.9) remained unchanged, while the systemic content of GMCSF and IL-2 was below the detectable threshold. Notably, the concentration of the anti-inflammatory cytokine IL-10 was downregulated by ultrasound stress (U = 1, *p* = 0.0159). For more details, please see Appendix A. Quantitative PCR analysis in the hippocampus showed that stressed females had overexpression of GSK-3β, a marker of distress and depressive syndrome (U = 12, *p* = 0.001, Figure 3B), along with increased expression of IL-1β and IL-6 (U = 17, *p* = 0.02 and U = 16, *p* = 0.017) compared to non-stressed females.

### 2.3. Female Offspring Born from Dams Exposed to Chronic Ultrasound or Challenged with LPS Share Behavioral Similarities

Adult female mice that were born in the chronic-stress study cohort demonstrated decreased sucrose preference (U = 7, *p* < 0.0001, Figure 4A), a change that was repeated in the immune challenge study (U = 28, *p* = 0.005, Figure 4A). In the three-chamber sociability test, in comparison with the respective controls, offspring from the chronic stress study showed a trend of shorter time spent interacting with the social stimulus mouse (U = 42, *p* = 0.100, Figure 4B) than was observed in the immune challenge study (U = 56, *p* = 0.219, Figure 4B). In the open field, mice from the chronic stress study exhibited decreased locomotor activity compared to the respective control group (U = 35, *p* = 0.038, Figure 4C), while this parameter remained unchanged in offspring from the immune challenge study (U = 73, *p* = 0.72, Figure 4C). Time spent in the center was significantly shorter in offspring from both the chronic stress (U = 0, *p* = 0.0001, Figure 4D) and immune challenge experiments (U = 39, *p* = 0.03, Figure 4D) compared to their respective control groups. In the elevated plus maze, animals from both the chronic stress and immune challenge experiments displayed a decreased number of open arm entries, while only the chronic stress cohort spent less time in the open arms (U = 0, *p* = 0.0001; U = 30, *p* = 0.018; U = 12, *p* = 0.0001 and U = 53, *p* = 0.159, respectively, in comparison with the relevant control groups, Figure 4E,F).

### 2.4. Chronic Ultrasound Exposure Induces Cognitive Deficits in the Female Offspring in the Barnes Maze

Repeated measures two-way ANOVA revealed a significant interaction between the day and treatment factors (F12,144 = 4.78, *p* < 0.01, repeated measures two-way ANOVA, Figure 5A). A significant decrease in the primary latency to escape over the five days of training was observed in the control groups only. In both control groups, a significant decrease in latency was observed between days 1, 2, 3 and day 5 (all *p* < 0.01, Tukey’s test). Additionally, a significant decrease was found between day 1 and day 4 in the animals from non-stressed dams (*p* = 0.04, Tukey’s test), and between day 4 and day 5 in animals from saline-injected dams (*p* = 0.02, Tukey’s test). Tukey’s post hoc test revealed significant group differences only on day 5, where mice from both LPS-challenged and ultrasound-exposed dams showed a significantly longer primary latency to escape compared to both saline-injected and non-stressed control groups (all *p* < 0.01, Tukey’s test). In the probe test on day 6, the offspring from the ultrasound-exposed dams had longer time to escape in comparison with the respective control (*p* < 0.01, unpaired *t*-test, Figure 5B), the effect that was got in the pups from LPS-injected dams compared to corresponding control as well (*p* < 0.01, unpaired *t*-test).

## 3. Discussion

The present study demonstrated that chronic exposure to unpredictably alternating ultrasound frequencies, that are naturally emitted by rodents to communicate with conspecifics, is capable of inducing depressive syndrome in female mice, associated with systemic pro-inflammatory changes, and eliciting behavioral deficits in offspring. The present data are in accordance with earlier demonstrated findings in males that 21 days of exposure of BALB/c mice to unpredictably alternating frequencies of ultrasound between the ranges of 20–25 and 25–45 Hz led to decreased sucrose intake, a sign of anhedonia in rodents and increased anxiety-like behavior, as shown by decreased time spent in the center of the open field and open arms in the elevated plus maze [29,37]. Notably, here we demonstrated that ultrasound exposure causes pro-inflammatory changes locally in the hippocampus of dams and systemically in the blood, suggesting a profound pro-inflammatory effect of ultrasound stress. It has recently been reported that hippocampal overexpression of IL-1β and IL-6 is associated with abnormal learning of adverse memories, a process that models the development of a depressive-like state in its early stages [40]. Moreover, hippocampal overexpression of these cytokines is linked to the mechanisms of vulnerability to the development of depressive syndrome [41]. Additionally, the overexpression of GSK-3β in the hippocampus, that we revealed in pregnant dams, is known to be a marker of distress and depressive syndrome [42]. In this paper, we report an over-expression of GSK-3β, IL-1β, and IL-6 mRNA in the hippocampus of ultrasound-exposed females, which is similar to the changes found in males [29,30,37].

Neuroinflammation, on its own, has long been recognized as a causative factor for emotional stress and psychiatric disorders, including depressive syndrome [40], and is regarded as a risk factor for the development of depressive syndrome [41,42]. The chronic systemic production of inflammatory cytokines, such as IL-1β and IL-6, is also closely related to the etiology of depressive syndrome [43,44]. Notably, extensive studies have demonstrated that maternal immune activation is a risk factor for the development of ASD-like abnormalities in offspring [45]. The developing central nervous system is particularly vulnerable to inflammatory insults [13], especially during critical windows, such as embryonic day 12.5 [13,14]. These perturbations affect the brain and behavior in an irreversible manner and persist throughout an animal’s life [13]. Here, we found increased concentrations of several pro-inflammatory cytokines, such as IL-1β, IL-6, IL-17, RANTES and TNFα, in the serum of female mice that were exposed to the ultrasound stress. These changes have been reported to contribute to the pathophysiology of both depressive syndrome [1] and the neurodevelopmental pathologies of exposed embryos [46,47,48].

For example, IL-1β and IL-6 have been shown to regulate neural stem-cell differentiation and sprouting and activation of resident microglia and astrocytes [45,46]. Studies with transgenic mouse lines have shown that the balance of these cytokines in an embryo is crucial for normal hippocampal and hypothalamic neurogenesis, and is linked to cognitive performance and social behavior [49,50,51]. Increased IL-17 and TNFα signaling has been reported to promote inflammation and oxidative stress in monocytes of autistic children and adults, which is a mechanism to maintain low-grade inflammation in the systemic circulation [52,53]. Moreover, a combination of high levels of blood concentrations IL-17, IL-1β, and TNFα was suggested to constitute a characteristic plasma cytokine profile of patients with ASD [11,17]. Among plasma chemokines, increased RANTES content is positively correlated with behavioral impairment of individuals with ASD [54,55]. Importantly, decreased blood IL-10 content, a key anti-inflammatory cytokine [56], in ultrasound exposed mice, that was detected in our study, further suggests an altered cytokine profile in these animals.

Here we demonstrated that the offspring of dams exposed to the chronic ultrasound stress or LPS immune challenge displayed aberrant emotional behavior and impaired learning. Both challenges resulted in pronounced depressive-like and anxiety-like changes in behavior, while no significant alterations in the sociability were found. The offspring of dams that were exposed to the chronic ultrasound stress or LPS showed compromised acquisition of spatial memories in the Barnes maze. Overall, the behavioral changes that were induced by either challenge in pregnant dams were comparable between the two groups of offspring. Hence, our data suggest the usefulness of ultrasound stress, as applied here, for modeling neurodevelopmental behavioral abnormalities that can likely mirror the negative effects of ‘emotional stress’ in pregnant women. It is remarkable that the behavioral changes induced by the ultrasound were comparable to those caused by robust immune stimulation in response to LPS [57,58]. Taken together, the results of the present study suggest the validity of the ultrasound stress paradigm of alternated frequencies for modeling maternal systemic inflammation, and this warrants further studies.

## 4. Materials and Methods

### 4.1. Animals

Experiments were performed on 35 female BALB/c mice that were 3 months old (adult females study part) and 92 female BALB/c that were 2 months old (offspring females study part); animals were provided by a provider licensed by Charles River (http://www.spfanimals.ru/about/providers/animals (accessed on 1 January 2018). Animals were maintained on a reversed 12 h light/dark cycle under controllable laboratory conditions (22 ± 1 °C, 55% humidity, room temperature 22 °C; lights were on at 21:00). Animals were single-housed in standard cages. Since single housing alone can cause distress in female mice, we undertook measures to compensate for this. For this propose, we housed all mice in an enriched environment using facial tissue and wooden objects. We also refrained from the use of ventilated cages so animals could be exposed to olfactory cues from each other. Food and water were available ad libitum. Where applicable, experimental groups were balanced by their age, body weight, and plug days. Weaning occurred on day 21 after birth.

### 4.2. LPS Challenge

Adult female mice (*n* = 5) received two intra-peritoneal (i.p.) injections of 75 µg/kg LPS (5/µLg lipopolysaccharide O127:B8; Sigma, Darmstadt, Germany, Cat. #L3129) given on both day E11.5 and day E12.5. A separate group of control female mice (LPS control) was injected with saline on days E11.5 and E12.5. Injection procedure was conducted as described elsewhere [33,59]. The first day when a plug was checked was counted as day E0.5.

### 4.3. Ultrasound Exposure 

Ultrasound, of a frequency range of 20–45 Hz and an average intensity of 50 dB, was constantly delivered in an acoustically isolated room using a random schedule of alternating frequencies for 21 days, via a manufactured device (Weitech, Wavre, Belgium), as previously described [29,30,31,38,39]. The loudness of the sound fluctuated in the range ±10% of the averaged value, i.e., ±5 dB. The range of ultrasound stimulation frequency alternated each 10 min between the following intervals: low frequencies (20–25 kHz), middle range frequencies (>25 < 40 kHz), and high range frequencies of (40–45 kHz). During the 10 min periods, ultrasound frequencies fluctuated at variable short time spans of ≤1 s (averaged frequency 70 Hz ± 10 Hz) at the above-indicated intervals. Low- and middle-frequency ultrasound radiation constituted 35% of the emission time while high frequencies constituted 30% of the emission time. The even distribution of ultrasound radiation was controlled by the use of an ultrasound detector (Discovery Channel, Silver Spring, MD, USA). Mice were housed individually in standard plastic cages (30 × 20 × 1 cm) during the ultrasound stress. The ultrasound device was hung 2 m above the cages of the experimental groups with an average horizontal distance between cages of 2.5 m. The position of the cages, with respect to the stimulator, was changed every two days. For more details, please see Appendix A.

### 4.4. Sucrose Test

During this test, for 24 h, mice were given a free choice between two bottles, one with a 1% sucrose solution and the other with tap water. Mice were tested immediately after ultrasound exposure. At the beginning and end of the test, the bottles were weighed and consumption was calculated. The beginning of the test started with the onset of the dark (active) phase of the animals’ cycle. To prevent the possible effects of side-preference in drinking behavior, the position of the bottles in the cage was switched after 12 h. No previous food or water deprivation was applied. The percentage of preference for sucrose was calculated using the following formula: sucrose preference = volume (sucrose solution)/(volume (sucrose solution) + volume (water)) × 100, as described elsewhere [35,36,60,61].

### 4.5. Open Field

The open field test was carried out in square boxes (45 × 45 × 45 cm) made from grey plastic and illuminated with white light (25 Lux). The experimental animal was placed near the wall and its movements were tracked for a 5 min period. Travel distance, a parameter of horizontal activity, was measured as the number of crossed squares. Time spent in the central area (15 × 15 cm) was scored as a parameter of anxiety-like behavior [30]. Previously reported studies with the settings of the open field test employed here suggest good reliability of selected parameters in measuring anxiety and locomotion [29,30,31,38,39,62].

### 4.6. Elevated plus Maze

The apparatus had two open arms (25 × 5 × 0.5 cm) perpendicular to two closed arms (25 × 5 × 16 cm) with a center platform (5 × 5 × 0.5 cm) and was made from grey plastic. Illumination was 75 Lux in the central platform and open arms, with no direct light to the closed arms [28,29,39].

### 4.7. Three-Chamber Sociability Test

A three-chambered apparatus (102 × 47 × 45 cm) with transparent plexiglass walls was used. Wire cups for social and non-social stimuli were 10 cm in diameter and 15 cm high. An age- and sex-matched unfamiliar mouse served as a social stimulus. A glass brick served as a non-social stimulus. The test was conducted under 50 Lux with the duration of 10 min. Three trials per animal were conducted. Time spent with social stimulus was scored [18].

### 4.8. Barnes Maze

The maze is an elevated, circular, light-grey platform D = 100 cm. A black chamber was hidden under one of the 20 holes around the perimeter, and the other 19 holes were left empty. Bright lights of 600 lx were used as a mild aversive stimulus, and four visual cues were placed on the walls around the maze. Each mouse had 5 acquisition days in the row with 3 trials per day. The 6th day was the probe trial [16].

### 4.9. Culling, Brain Dissection, Serum Collection, RNA Extraction, and Quantitative RT-PCR

Mice from each experimental group were anaesthetized using isoflurane, as described elsewhere [30,31,37]. The left ventricle was perfused in situ with 10 mL ice-cold saline. Blood collection was performed transcardially and blood was stored in heparinized vials prior to centrifugation (1500 rcf, 15 min, 4 °C); 100 μL of plasma was removed and immediately stored at −20 °C until use, as described elsewhere [29,30].

The brain of each mouse was dissected and the hippocampus was isolated as previously reported [31,37] and quickly placed in RNAlater solution and stored at +4 °C for a few days, until used for extraction of total RNA. mRNA was extracted by using TRI Reagent. The concentration of total RNA in the obtained samples was measured using Nanovue Plus (Biochrom Ltd., Cambridge, UK). Reverse transcription was performed using a set by “EUROGEN” on a 2720 thermal cycler. Then, 1 μg total RNA was converted into cDNA using deoxyribonuclease I. Quantitative RT-PCR (qRT-PCR) was performed on a Step One Plus thermal cycler. Data were normalized to GAPDH mRNA expression and calculated as relative-fold changes compared to control rats, as described elsewhere [31,37]. Primers are listed in the Appendix A for the following targets: GAPDH, GSK-3β, IL-1β, and IL-6. Each experimental sample was measured in triplicate. Results of qRT-PCR measurement were expressed as Ct values, where Ct is defined as the threshold cycle of PCR at which the amplified product was 0.05% of the normalized maximal signal. We used the comparative Ct method and computed the difference between the expression of the gene of interest and GAPDH expression in each cDNA sample (2^−ΔΔCt^ method). Data are given as expression-folds compared to the mean expression values in control mice [29,31].

### 4.10. Quansys Q-PlexTM ELISA Array

We used 7–8 mice that were randomly selected from the cohorts of animals that were previously studied for behavior, for a subsequent ELISA analysis of serum cytokines and culled as described above (p. 4.9). Blood collection was carried out at sacrifice, which took place simultaneously in all mice, i.e., four days after the termination of ultrasonic stress exposure, on the day after behavioral tests were completed. Mouse cytokine 16-plex strip wells were used to detect GMCSF, IFNγ, IL-1α, IL-1β, IL-2, IL-3, IL4, IL-5, IL-6, IL-10, IL-12p70, IL-17, MCP-1, MIP-1α, RANTES, and TNFα. Serum was collected by decapitation of animals and stored at −20 °C before use. A sample volume per well was 30 µm. Serum was were centrifuged at 13,000× *g* for 10 min at 4 °C and then processed following the manufacturer’s instructions. A Q-View Imager (Quansys Biosciences, West Logan, UT, USA) was used to capture absorbance intensities using chemiluminescence. Data analysis for each cytokine expression was performed with Q-View software with data expressed with respect to the control group.

### 4.11. Statistical Analysis

GraphPad Prism software version 5.03 was used for data analysis. Comparison of the two groups was carried out using the Mann–Whitney test, since Kolmogorov–Smirnov did not show normal data distribution, except the with the data from Barnes maze. Repeated measures data from the Barnes maze test were assessed using repeated measures two-way ANOVA with post-hoc Tukey’s multiple comparisons test and as far as sphericity was not assumed, the Geisser-Greenhouse correction was applied. Statistical significance was set at *p* < 0.05.

## Figures and Tables

**Figure 1 ijms-24-11712-f001:**
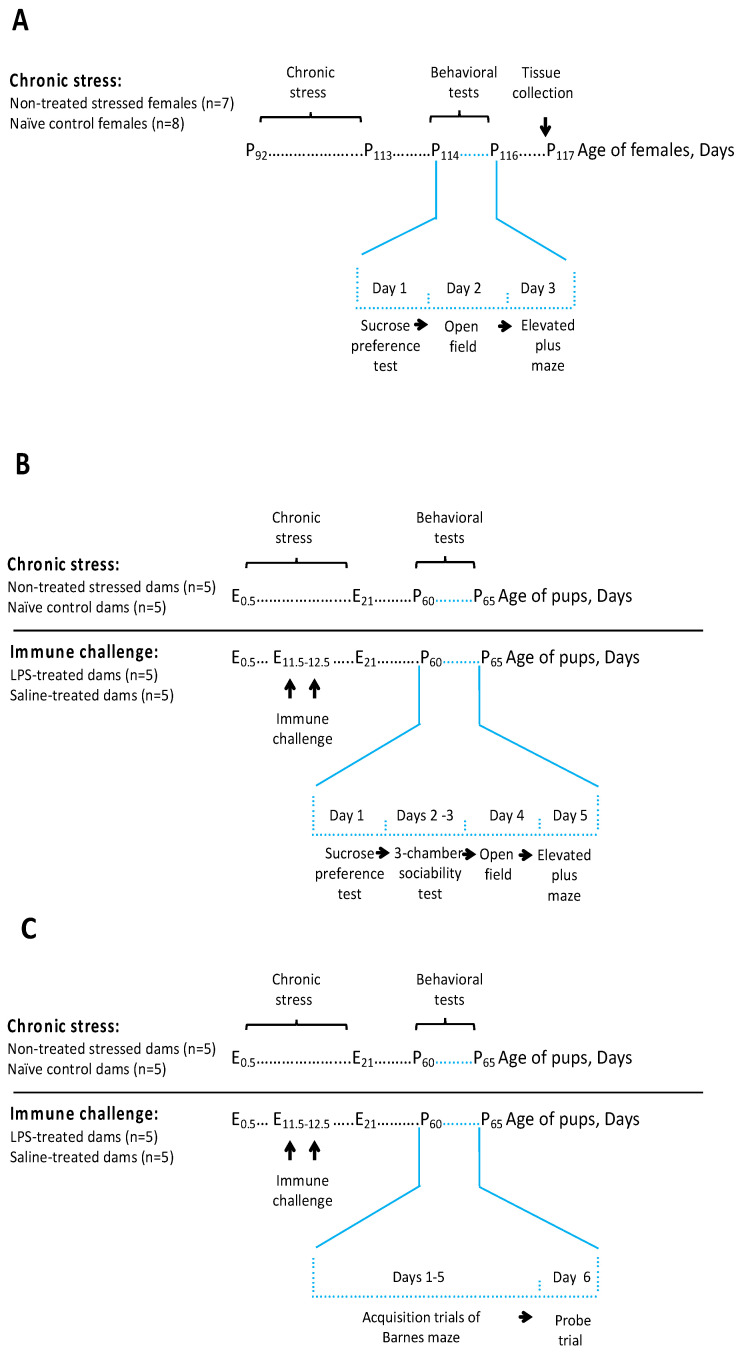
Experimental scheme. Virgin females at the age of 3 months (average age was postnatal day P92) were exposed to 21 days of ultrasound of alternating frequencies (**A**). The sucrose preference test, open field test, and elevated plus maze were used to examine behavioral changes in adult females that had been exposed to ultrasound stress. Tissue collection was conducted after behavioral tests for biochemical analysis the next day. Next, we studied behavioral changes in the offspring of ultrasound-exposed dams starting at day P60. These effects were compared against the effects of double LPS administration on the offspring of pregnant dams at embryonic days E11.5 and E12.5, in two separate runs (**B**) using a battery of tests for emotionality: the sucrose preference test, open field test, and elevated plus maze and (**C**) the Barnes maze. Chronic ultrasound exposure started the same day that the plugs were checked; plug day was set as day E0.5. There were 10 offspring controls in the studies (**B**,**C**), and 14 and 16 offspring mice in the chronic stress and immune challenge studies, respectively.

**Figure 2 ijms-24-11712-f002:**
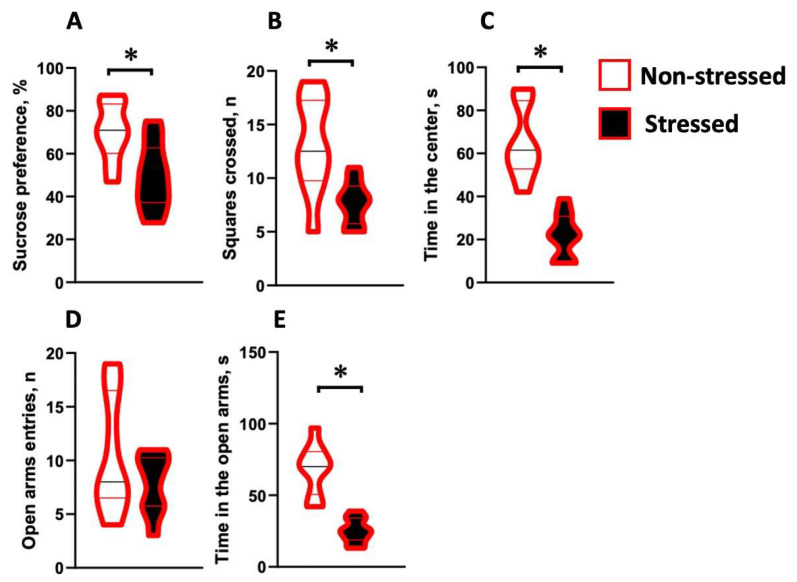
Chronic ultrasound stress induced a depressive-like syndrome and elevated anxiety-like behavior in adult female mice. In the sucrose preference test (**A**), stressed mice had a lower preference to sucrose than the control animals. In the open field test, ultrasound exposure decreased locomotor activity (**B**) and time spent in the center (**C**) compared to control animals. In the elevated plus maze, the number of open-arm entries (**D**) remained unaffected by the stress exposure, while time spent in the open arms (**E**) decreased in the stressed animals compared to the non-stressed group. Each group had 5 animals. * *p* < 0.05, Mann–Whitney test.

**Figure 3 ijms-24-11712-f003:**
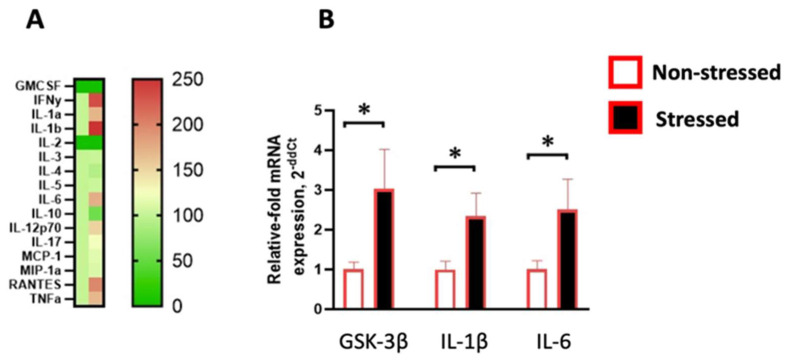
Chronic ultrasound stress induced pro-inflammatory changes in the systemic circulation and hippocampus of females. The heat map (**A**) represents the concentration of key cytokines, including GMCSF, IFNγ, IL-1α, IL-1β, IL-2, IL-3, IL-4, IL-5, IL-6, IL-10, IL-12p70, IL-17, MCP-1, MIP-1α, RANTES, and TNFα in the systemic circulation of stressed animals (second column) normalized to the control group (first column). Relative data are displayed as % change from the control level. Zero values (dark green) represent cytokines with undetected concentrations. mRNA expression levels (**B**) of GSK-3β, IL-1β, and IL-6 were upregulated in stressed females compared to control mice. * *p* < 0.05, Mann–Whitney test. Bars are mean ± SEM.

**Figure 4 ijms-24-11712-f004:**
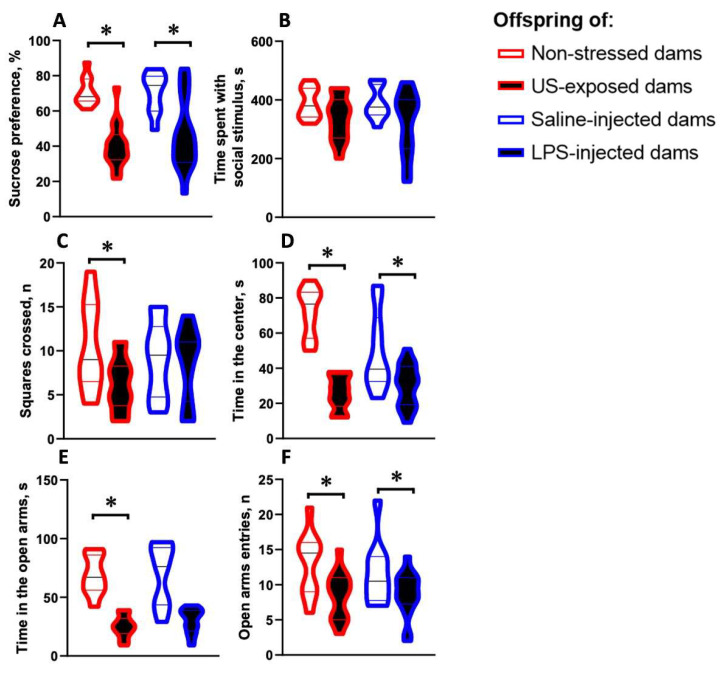
Depressive- and anxiety-like changes in the offspring of dams challenged with the ultrasound or LPS. In the sucrose preference test (**A**), we observed lower sucrose preference in the offspring born from ultrasound-exposed or LPS-challenged dams than in their respective controls. In the three-chamber sociability test (**B**) there was a trend towards reduced time spent with the social stimulus in the offspring of ultrasound-exposed dams compared to the control groups (see the text); no other group differences were found. In the open field test, the offspring from the ultrasound stress experiment displayed decreased locomotor activity (**C**), while no such changes were observed in the immune challenged group. Time spent in the center of the open field (**D**) was decreased in the offspring from both studies compared to the control groups. In the elevated plus maze, in comparison to control mice, the number of open-arm entries (**E**) and time spent in the open arms (**F**) were decreased in the offspring of challenged dams. Behavioral data are presented for the 10 control female offspring and 14 offspring from dams exposed to ultrasound, and from 10 control female offspring and 16 offspring from dams exposed to LPS. * *p* < 0.05, Mann–Whitney test.

**Figure 5 ijms-24-11712-f005:**
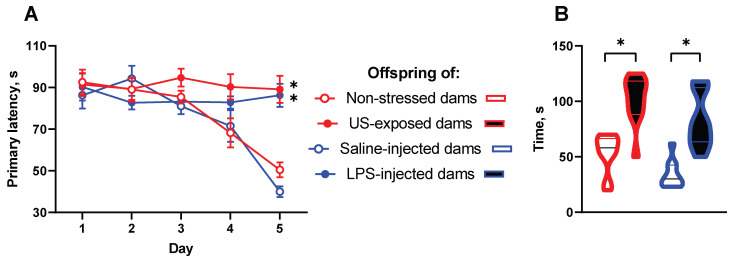
Chronic ultrasound stress and LPS challenge of pregnant dams compromised memory acquisition in the Barnes maze by the offspring. (**A**) Over the five-day acquisition period, a significant decrease in the primary latency to escape was observed in the control groups only. The primary latency to escape in the offspring born from ultrasound-exposed dams and LPS-challenged dams was significantly longer than in the control on day 5. (**B**) In the probe test, both ultrasound-exposed and LPS-challenged offspring revealed a significantly prolonged time to find the hidden box. The group composition included 10 control female offspring and 12 offspring from dams exposed to ultrasound, and 10 control female offspring and 13 offspring from dams exposed to LPS. * *p* < 0.05 (**A**) repeated measures two-way ANOVA and Tukey’s test. Only group differences are indicated in the figure. (**B**) Unpaired *t*-test.

## Data Availability

The data can be requested on demand.

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
