# Peer review of "Maternal Chronic Ultrasound Stress Provokes Immune Activation and Behavioral Deficits in the Offspring: A Mouse Model of Neurodevelopmental Pathology"

_ijms, 2023, doi:10.3390/ijms241411712_

Round 1

Reviewer 1 Report

Comments to the Author

Comments:
Manuscript ID: 2385329, Title: Maternal chronic ultrasound stress exerts immune activation and behavioural abnormalities in the offspring: a mouse model of neurodevelopmental pathology

The authors did an interesting and very good job on the effects of stress (ultrasound) on mothers and the biochemical and behavioral repercussions of their offspring. The study presents an integrative approach, evaluating biochemical aspects in the brain and blood and also behavioral responses to different situations and tests. Although the study is well written, it is necessary for the authors to review some questions and suggestions that I present below.

Title: I suggest authors change the word "abnormalities" in the title or write the title in another way.

Keywords: Please avoid repeating words that already exist in the title. You can use synonyms, but not repeat the same ones.

L70: Define the acronym LPS and then you can continue using the acronym.

L111: Just here they define LPS, but I had already mentioned it before.

L107-110: What do they mean by behavioral, molecular and biochemical abnormalities? If you mean: "gene expression of markers of inflammation, plasticity and cellular distress", isn't this what you commented in the first part of the sentence? If so, I suggest rewriting this sentence so that it is not repetitive.

L:108 “behavioral” and L2: “behavioural”, choose a way of writing English, either American or British, but not both.

L336-337: This sentence “Mice were housed individually in standard plastic cages (30x20x15cm) during the ultrasound stress part.” I suggest removing it from this paragraph, since there is another paragraph linked to the exposure of stressors.

L342-364: Add the "n" of animals used in each group, either mothers or their pups and either with the ultrasound stressor or with LPS.

L348-364: What was the total exposure time to the ultrasound stressor for each animal?

L376-381: Why were other types of behavioral indicators not evaluated in this test, for example, rearing, number of feces, among others?

Why did the authors decide to use 5 min in this test and not 10 min?

L376-385: I suggest including references to these tests.

L399: How were the 5 animals of each group chosen?

L401: Why did the authors dissect the prefrontal cortex of the brain if they do not show any results from this area? In the event that results from this region of the brain are presented, I suggest that the authors highlight it clearly in the results and separately from those of the hippocampus.

L418-425: All these cytokines were measured from blood samples? If they are measured from blood samples, the process for obtaining the respective samples must be described. This includes describing the clamping process, vein of blood collection, how many mL obtained per animal, were cytokines measured in serum or plasma? If it is plasma, was it with what anticoagulant? At what experimental point were the blood samples obtained? It is not the same if it is carried out immediately after a stressor than if it is carried out in basal conditions or at sacrifice.

L424: Mention that "data normalization" is not a correct form, since normalization is linked to the distribution of the variables. For this reason, I suggest that the authors change to "expressed with respect to the control group".

L427-428: Why didn't the authors use the Shapiro-Wilk test? The Kolmogorov-Smirnov test is not recommended for small sample sizes, as seems to be the case in this study.

Did the authors perform sample size calculation tests?

Did the authors assess the statistical power of their results and the size of the effect? It is very important that authors review these concepts when making inferences.

Figure 5 A. If the same animals were used for the latency records at different times (days), then the comparisons between groups for a single day in a totally separate way is not the best or most correct way to analyze these results. I suggest the authors review the analysis of generalized linear models, in which the distribution of the variables can also be included. The new analyzes will allow the authors to assess overall group effects, between stressed and unstressed by stress condition, but also to assess interactions between group effects and days. In addition, since the same animals are repeated in days, the error of the repetitions is reduced if an analysis of repeated measures is carried out using linear models, which gives more support to the data, beyond being really correct.

Figure 5 A: Missing legend on x-axis.

Why didn't the authors do confocal analysis studies to evaluate molecules in the hippocampal regions, or to see if the dentate gyrus (site of neurogenesis) was affected in the pups?

Neither were proteins or metabolic pathways that could be affected in the hippocampus analyzed? This would be interesting to raise in the discussion of the work, since it is not known how these stressful conditions affect the mother and her pups at a central molecular level in the brain.

What sex were the pups? If there were of both sexes, how many were of each?

L:108 “behavioral” and L2: “behavioural”, choose a way of writing English, either American or British, but not both.

Author Response

Comments Reviewer 1

The authors did an interesting and very good job on the effects of stress (ultrasound) on mothers and the biochemical and behavioral repercussions of their offspring. The study presents an integrative approach, evaluating biochemical aspects in the brain and blood and also behavioral responses to different situations and tests. Although the study is well written, it is necessary for the authors to review some questions and suggestions that I present below.

We are grateful for the positive comments and very helpful recommendations expressed by Reviewer 1. All the recommendations of this reviewer were implemented in the manuscript. As requested, we address methodological questions that were not presented or explained with sufficient clarity, in the text now. We would like to thank Reviewer 1 for bringing to our attention the inconsistences and flaws in the description of the Methods, data presentation, stylistic problems in the ms text that are fixed now. In particular, we like to thank Reviewer 1 for a suggestion to use more correct statistical methods. As recommended, the data from Barnes maze experiments performed on offspring mice are now analysed with repeated two-way ANOVA, respective changes were made in the Methods, Results and Figure legends. As recommended, we include now the group sizes in a more clear way. The most part of questions rose by this Reviewer were addressed in the ms text, while some were commented in this Response letter. New references were added. Together, we feel that the suggestions and questions of Reviewer 1 have substantially contributed in the improvement of our work and would like to express our gratitude for their important input.

 1.Title: I suggest authors change the word "abnormalities" in the title or write the title in another way.

We thank the reviewer 1 for this suggestion, as recommended, we have revised ms title now.

The former title

“Maternal chronic ultrasound stress exerts immune activation and behavioural abnormalities in the offspring: a mouse model of neurodevelopmental pathology”

is replaced with a new Title:

“Maternal chronic ultrasound stress exerts immune activation and behavioural deficits in the offspring: a mouse model of neurodevelopmental pathology”

2.Keywords: Please avoid repeating words that already exist in the title. You can use synonyms, but not repeat the same ones.

Following this recommendation of Reviewer 1, we have replaced former keywords:

“ultrasound stress; LPS; inflammation; interleukins; depressive syndrome; developmental abnormalities; mice”

with new keywords:

 “ultrasound radiation; systemic inflammation; pro-inflammatory cytokines; depression; memory; offspring; mice”

  1. L70: Define the acronym LPS and then you can continue using the acronym.

We apologize for this error, it was fixed now, these changes are implemented throughout the text.

4.L111: Just here they define LPS, but I had already mentioned it before.

Thank you, these changes are implemented throughout the text.

5.L107-110: What do they mean by behavioral, molecular and biochemical abnormalities? If you mean: "gene expression of markers of inflammation, plasticity and cellular distress", isn't this what you commented in the first part of the sentence? If so, I suggest rewriting this sentence so that it is not repetitive. 

We are grateful to Reviewer 1 for pointing out that above-mentioned sentence sounds repetitive Indeed, this part of the text was extensively revised and extended, in response to recommendations of Reviewer 2.

6.L:108 “behavioral” and L2: “behavioural”, choose a way of writing English, either American or British, but not both.

 Thank you for bringing these inconsistencies to our attention, we made sure that in the ms text the British spelling is consistently used now.

7.L336-337: This sentence “Mice were housed individually in standard plastic cages (30x20x15cm) during the ultrasound stress part.” I suggest removing it from this paragraph, since there is another paragraph linked to the exposure of stressors.

In response to this suggestion, we have moved above-indicated sentence to the recommended part of the Methods section 4.3 “Ultrasound exposure”.

  1. L342-364: Add the "n" of animals used in each group, either mothers or their pups and either with the ultrasound stressor or with LPS

We thank Reviewer 1 for this suggestion and now in addition to group sizes indicated in Figure 1 and Materials and Methods section, we added the "n" of all groups in Figure legends.

9.L348-364: What was the total exposure time to the ultrasound stressor for each animal?

The animals were subjected to ultrasound of alternating frequencies for 21 day. We made sure that this detail is clearly indicated throughout the text.

  1. L376-381: Why were other types of behavioral indicators not evaluated in this test, for example, rearing, number of feces, among others? Why did the authors decide to use 5 min in this test and not 10 min?

We appreciate that larger number of behavioural parameters can be registered in the open field, besides those that were scored here, i.e., distance travelled and time spent in the central area of the arena. We agree that rearing activity and number of feces, among others, are often used as valuable read-outs of the open field behaviour in rodents. However we chose to relay on the read-outs in the open field test that previously were validated in similar experiments and employed here settings (Morozova et al., 2016; Pavlov et al., 2019; Sambon et al., 2020; Strekalova et al., 2018). Indeed, distance travelled and time spent in the central area were shown to accompany altered metabolic and biochemical hallmarks in stressed mice, such as  elevated blood levels of CORT (Pavlov et al., 2019), increased expression of stress marker cFOS and BDNF in key limbic structures (Costa-Nunes et al., 2020; Gorlova et al., 2019), overexpression of oxidative stress markers in key limbic structures (Gorlova et al., 2019; Pavlov et al., 2019). Finally, chronic administration of classic antidepressant therapy was shown to ameliorate open field behaviour in mice exposed to chronic ultrasound stress and studied under employed here settings (Morozova et al., 2016).

Besides, in many of our previous studies we have used 5 min-open field test as a reliable behavioural tool in situations, where animals are anticipated to display marketable group differences, e.g., in chronic stress experiments. Again, a protocol with short duration of the open test was extensively validated in our previous experiments. We fully agree with Reviewer 1 that it would be potentially beneficial to prolong a period of recordings in this test. However, due to large number of animals and a need to preclude the influence of serious confounds associated with circadian factor in mice, it is necessary to limit the duration of open field recordings by a 5-min period. In addition, earlier reported studies showed good validity and reliability of a 5-min scoring of selected behavioural measures in our settings.

For better clarity and justification of the selection of open field parameters to be studied in our work, we have introduced a new comment in the Methodology section:

 “Previously reported studies with employed here settings of the open field test suggest good reliability of selected here parameters in measuring anxiety and locomotion (Costa-Nunes et al., 2020; Gorlova et al., 2019; Morozova et al., 2016; Pavlov et al., 2019; Sambon et al., 2020; Strekalova et al., 2018)”.

 11 L376-385: I suggest including references to these tests.

Following this suggestion of Reviewer 1 we have added the references in the description of behavioural paradigms.

12 L399: How were the 5 animals of each group chosen?

In response to this question, we have added an explanation of how were the 5 animals of each group chosen, in the ms text, Materials and Methods section to p.4.9:

Five mice of each experimental group were randomly selected from total number of animals that were anaesthetized using isoflurane as described elsewhere”.

  1. L401: Why did the authors dissect the prefrontal cortex of the brain if they do not show any results from this area? In the event that results from this region of the brain are presented, I suggest that the authors highlight it clearly in the results and separately from those of the hippocampus.

We apologize for this typo, but no samples from the prefrontal cortex were used in this study, respective corrections are made in the text.

  1. L418-425: All these cytokines were measured from blood samples? If they are measured from blood samples, the process for obtaining the respective samples must be described. This includes describing the clamping process, vein of blood collection, how many mL obtained per animal, were cytokines measured in serum or plasma? If it is plasma, was it with what anticoagulant? At what experimental point were the blood samples obtained? It is not the same if it is carried out immediately after a stressor than if it is carried out in basal conditions or at sacrifice.

We thank reviewer 1 for bringing to our attention this flaw in the description of the methodology, now this part is added to the Materials and Methods part, respective changes were made in the subheading of the section 4.9 and the first para of this section:

“The left ventricle was perfused in situ with 10 mL ice-cold saline. Blood collection was performed transcardially, blood was stored in heparinized vials prior to centrifugation (1500 rcf, 15 min, 4 °C); 100 μl of serum was removed and immediately stored at − 20 °C until use as described elsewhere.”.

We used 7-8 mice that were randomly selected from the same cohorts of animals that were studied for behaviour, for a subsequent ELISA analysis of serum cytokines. Therefore, blood collection was carried out at sacrifice that took place simultaneously in all mice, on the next day after behavioural tests were finished, i.e. this was done four days after the termination of ultrasonic stress exposure. This information is now added to the ms text.

  1. L424: Mention that "data normalization" is not a correct form, since normalization is linked to the distribution of the variables. For this reason, I suggest that the authors change to "expressed with respect to the control group".

Following this suggestion, we have edited above-mentioned part of the Methods as recommended, using the sentence "expressed with respect to the control group".

  1.  L427-428: Why didn't the authors use the Shapiro-Wilk test? The Kolmogorov-Smirnov test is not recommended for small sample sizes, as seems to be the case in this study. 

We agree with this criticism. The Kolmogorov-Smirnov is widely and traditionally used in works like ours and can be used for the sample sizes we have in our study, but Shapiro-Wilk is indeed preferable due to its higher power (Ghasemi and Zahediasl, 2012). To answer this criticism, we have run the Shapiro-Wilk test on our data which revealed a similar outcome (data not shown).

  1. Did the authors perform sample size calculation tests?

Since we have based a study design selected for this work on our previous experience with the ultrasound stress model (Costa-Nunes et al., 2020; Gorlova et al., 2019; Pavlov et al., 2019) and LPS challenge (Sambon et al., 2021; Schapovalova et al., 2022; Strekalova et al., 2021) we have not carried out separate sample size determination for the purpose of this work. See also p. 17 of comments to Reviewer 2 of this Response letter.

  1. Did the authors assess the statistical power of their results and the size of the effect? It is very important that authors review these concepts when making inferences.

We fully agree with this view of Reviewer 1 that it can be very important to assess the statistical power of the results and the size of the effect. However, because non-significant results were not interpreted in this study, and probabilistic statistical methods with appropriate significance level (α=0.05) were used along with multiple comparisons correction, statistical power analysis could be omitted.

  1. Figure 5 A. If the same animals were used for the latency records at different times (days), then the comparisons between groups for a single day in a totally separate way is not the best or most correct way to analyze these results. I suggest the authors review the analysis of generalized linear models, in which the distribution of the variables can also be included. The new analyzes will allow the authors to assess overall group effects, between stressed and unstressed by stress condition, but also to assess interactions between group effects and days. 18. In addition, since the same animals are repeated in days, the error of the repetitions is reduced if an analysis of repeated measures is carried out using linear models, which gives more support to the data, beyond being really correct.

We greatly appreciate this important suggestion of Reviewer 1 that was full accepted. As recommended, we have treated the data obtained in the Barnes maze with repeated two-way ANOVA. Meanwhile, because there was no single factor difference between the control groups, it was not 2×2 study design, and thus, a tree-way repeated measures ANOVA cannot be applied for these data.

Respective changes were made to the p. 4.11, Statistical analysis section in the “Materials and methods” section:

Repeated measures data from the Barnes maze test was assessed using repeated measures two-way ANOVA withpost-hoc Tukey’s multiple comparisons test. As far as sphericity was not assumed, Geisser-Greenhouse's correction was applied”.

As well as in Results section, the former description of the data from the Barnes maze was replaced with the following text:

Repeated measures two-way ANOVA revealed a significant interaction of day and treatment factors (F­12,144=4.78, p<0.01, repeated measures two-way ANOVA). Post-hoc Tukey’s test revealed significant group differences only on day 5, where mice from both LPS-treated and stressed dams, showed significantly increased primary latency to escape compared to both saline-injected and non-stressed control groups (all p<0.01, Tukey’s test). In addition, significant decreased of the primary latency to escape over five days of training was observed in control groups only. In both control groups, significant decrease in latency was observed between both days 1, 2, 3 and day 5 (all p<0.01, Tukey’s test). Additionally, significant decrease was found between day 1 and day 4 in the animals from non-stressed mothers (p=0.04, Tukey’s test), and between day 4 and day 5 in animals from saline-injected dams (p=0.02, Tukey’s test).”

  1. Figure 5 A: Missing legend on x-axis.

The legend has been fixed, thank you.

  1. Why didn't the authors do confocal analysis studies to evaluate molecules in the hippocampal regions, or to see if the dentate gyrus (site of neurogenesis) was affected in the pups?

We are grateful to Reviewer 1 for a suggestion to perform confocal microscopy on the hippocampus of the pups and will follow this advice in our follow-up studies. We feel that this work can be a large independent study that would need dedicated attention.

  1. Neither were proteins or metabolic pathways that could be affected in the hippocampus analyzed? This would be interesting to raise in the discussion of the work, since it is not known how these stressful conditions affect the mother and her pups at a central molecular level in the brain.

Again, thank you for this suggestion, we plan to study protein and metabolic pathways of the hippocampus of challenged females in the near future, as this is an important question that needs separate set of experiments.

23 What sex were the pups? If there were of both sexes, how many were of each?

We chose to run the present work exclusively on females so the behavioural results could be comparable between adult females and the pups. Pups born from experimental groups of females were of both sexes but unfortunately the exact number of each per group is difficult to retrieve.  

 References

Costa-Nunes, J.P., Gorlova, A., Pavlov, D., Cespuglio, R., Gorovaya, A., Proshin, A., Umriukhin, A., Ponomarev, E.D., Kalueff, A. V., Strekalova, T., Schroeter, C.A., 2020. Ultrasound stress compromises the correlates of emotional-like states and brain AMPAR expression in mice: effects of antioxidant and anti-inflammatory herbal treatment. Stress 23, 481–495. https://doi.org/10.1080/10253890.2019.1709435

Ghasemi A, Zahediasl S. Normality tests for statistical analysis: a guide for non-statisticians. Int J Endocrinol Metab. 2012 Spring;10(2):486-9. doi: 10.5812/ijem.3505.

Gorlova, A., Pavlov, D., Zubkov, E., Zorkina, Y., Inozemtsev, A., Morozova, A., Chekhonin, V., 2019. Alteration of oxidative stress markers and behavior of rats in a novel model of depression. Acta Neurobiol. Exp. (Wars). 79, 232–237. https://doi.org/10.21307/ane-2019-021

Morozova, A., Zubkov, E., Strekalova, T., Kekelidze, Z., Storozeva, Z., Schroeter, C.A., Bazhenova, N., Lesch, K.P., Cline, B.H., Chekhonin, V., 2016. Ultrasound of alternating frequencies and variable emotional impact evokes depressive syndrome in mice and rats. Prog. Neuro-Psychopharmacology Biol. Psychiatry 68, 52–63. https://doi.org/10.1016/j.pnpbp.2016.03.003

Pavlov, D., Bettendorff, L., Gorlova, A., Olkhovik, A., Kalueff, A. V., Ponomarev, E.D., Inozemtsev, A., Chekhonin, V., Lesсh, K.P., Anthony, D.C., Strekalova, T., 2019. Neuroinflammation and aberrant hippocampal plasticity in a mouse model of emotional stress evoked by exposure to ultrasound of alternating frequencies. Prog. Neuro-Psychopharmacology Biol. Psychiatry 90, 104–116. https://doi.org/10.1016/j.pnpbp.2018.11.014

Sambon, M., Gorlova, A., Demelenne, A., Alhama-Riba, J., Coumans, B., Lakaye, B., Wins, P., Fillet, M., Anthony, D.C., Strekalova, T., Bettendorff, L., 2020. Dibenzoylthiamine has powerful antioxidant and anti-inflammatory properties in cultured cells and in mouse models of stress and neurodegeneration. Biomedicines 8. https://doi.org/10.3390/BIOMEDICINES8090361

Sambon, M., Wins, P., Bettendorff, L., 2021. Neuroprotective Effects of Thiamine and Precursors with Higher Bioavailability: Focus on Benfotiamine and Dibenzoylthiamine. Int. J. Mol. Sci. 22. https://doi.org/10.3390/IJMS22115418

Schapovalova, O., Gorlova, A., de Munter, J., Sheveleva, E., Eropkin, M., Gorbunov, N., Sicker, M., Umriukhin, A., Lyubchyk, S., Lesch, K.-P., Strekalova, T., Schroeter, C.A., 2022. Immunomodulatory effects of new phytotherapy on human macrophages and TLR4- and TLR7/8-mediated viral-like inflammation in mice.

Strekalova, T., Bahzenova, N., Trofimov, A., Schmitt-Böhrer, A.G., Markova, N., Grigoriev, V., Zamoyski, V., Serkova, T., Redkozubova, O., Vinogradova, D., Umriukhin, A., Fisenko, V., Lillesaar, C., Shevtsova, E., Sokolov, V., Aksinenko, A., Lesch, K.P., Bachurin, S., 2018. Pro-neurogenic, memory-enhancing and anti-stress effects of DF302, a novel fluorine gamma-carboline derivative with multi-target mechanism of action. Mol. Neurobiol. 55, 335–349. https://doi.org/10.1007/s12035-017-0745-6

Strekalova, T., Svirin, E., Waider, J., Gorlova, A., Cespuglio, R., Kalueff, A., Pomytkin, I., Schmitt-Boehrer, A.G., Lesch, K.P., Anthony, D.C., 2021. Altered behaviour, dopamine and norepinephrine regulation in stressed mice heterozygous in TPH2 gene. Prog. Neuro-Psychopharmacology Biol. Psychiatry 108. https://doi.org/10.1016/j.pnpbp.2020.110155

Reviewer 2 Report

Abstract:  See comments regarding editing.

Introduction:  See comments regarding editing.  The stress resulting from ultrasonic frequencies being considered 'emotional' stress needs further elaboration.  Are there any specific behavior-associated noise that the frequencies were attempting to replicate, ex. a conspecific alerting to nearby predators?  Use of the word 'emotion' needs to be used with many qualifiers when working with animals.  While the previous study only used male mice, that is not enough justification to only use female offspring in this study, as the previous study did not examine males following in utero exposure.  As the present paper presents a different paradigm, both sexes should be used.  No direct comparison can be made between the current study and the previous, male-only, study.

Methods:  See comments regarding editing.  Mice being housed individually during ultrasound stress part - does this mean only during the ultrasound stress application?  Does it mean throughout all days within the time frame of the ultrasound stress?  Was anything done to compensate/take into account the stress that comes from being singly housed?  Were there any control, non-injected animals?  Injections, even of saline, give the animals considerable stress in and of themselves.  As this paper is intending to examine the result of emotional stress from very specific sources, care needs to be taken to reduce and control for stress from other sources.  Open field test - mice are prey animals; they prefer to be as sheltered as possible, meaning that the more anxious mouse would spend more time around the edge of the open field, while the less anxious mouse is more likely to spend time in the central area.  What parameters were scored on the elevated plus maze?  Section 4.9 describes procedures for both mouse and rat (line 401).

Results:  See comments regarding editing.  Figure 1 is not mentioned in the text.  The n's for groups are both small and substantially different.  Where is the paradigm for the offspring?  Figure 1 suggests there were three sets - ultrasound exposure, pregnancy and ultrasound exposure undergoing one set of behavioral tests, pregnancy and ultrasound exposure with a different set of behavioral tests.  Description within methods does not align with this figure.  Additionally, why were there no immune challenges in group A?  What was the point of group A?  Neither the chronic stress nor the behavioral tests, which were different than either groups B or C, were conducted at the same time as the other groups?  Was there any analysis on pregnancy loss/litter size changes due to stressful experiences during pregnancy?

"The top 10 females with the most pronounced deficits were selected for data analysis."  This statement requires justification - was there a scientific, statistically relevant reason for removing certain animals?  How many animals were removed?  What were the criteria for 'most pronounced deficits'.  Until or unless this is sufficiently explained, no results reported in this paper can be considered an accurate reflection of the experiments performed.

Section 2.1 - why is decreased sucrose preference considered a behavioral deficit?  In this section, time spent in the center was reduced, which is listed as an anxiety-like behavior, which is not the same as is described in other sections.  Figure 2, which links to section 2.1, mentions depressive-like syndrome, which is different from the anxiety-like behavior referenced in section 2.1.

Section 2.2 - were the animals used for this section the same reduced number that was used for the behavioral tests?  Were all animals used?  Were a different subset used?

Sections 2.3, 2.4 - See the aforementioned comments regarding using only the top 10 different females.  Why is it sometimes 'behavioral deficits' and sometimes 'cognitive deficits'?

Figure 5 - what was cut off from the top of Figure 5A?  This figure would be better represented with two sided error bars.

Discussion:  Conclusions are overstated due to previously discussed lack of reasoning for artificially limiting the sample size post data acquisition.  See previous comments regarding editing.  Consistency is needed between mentioning 'depressive' and 'anxiety' with regards to behavioral changes.  Again, decreased time in center of open field is considered a sign of increased anxiety-like behavior, in contrast to the methods section.  Is this paper intended to model depression, anxiety, ASD, neurodevelopmental disorders as a whole, or something else entirely?

"Some offspring are resilient to certain environmental challenges" - this is also seen in humans.  As a result, removing these animals from data analysis prevents any results from these experiments from being extrapolated to humans.

Several areas of the paper are lacking in grammar, most often with regards to lack of particles and articles.  An example is line 18, in the abstract.  'Major limitation' should be 'A major limitation'.  This lack of connecting words is seen throughout the paper, indicating a need for grammatical editing.  Editing is also needed for consistency of word choices; ex. dams vs. moms, behaviour vs. behavior.

Author Response

Reviewer 2

We would like to thank Reviewer 2 for their constructive criticism to our work and valuable suggestions to improve the manuscript. In particular, we highly appreciate a recommendation of this Reviewer to provide all behavioral data for the cohorts of the offspring that is now addressed in the paper. Following this recommendation, Figs. 3, 4 and 5 were edited, as well as respective results description, their discussion and Figure legends. In response to a recommendation of this reviewer, we have elaborated a bit more on the terms ‘emotional stress’ and ‘emotions’ in a context of animal models, and relayed the use of chronic ultrasound exposure of the frequencies applied here with regard to potency to induce ‘emotional stress’ and a depressive-like state, in the Introduction. In the revised paper, the Materials and Methods section, we provide a detailed description of the used ultrasound protocol and have made it clearer that its properties can mimic natural vocalizations in rodents. Also, we have added an explanation for the choice of behavioural protocols and study design, and indicated group sizes throughout the ms text and Figure legends for better clarity. Following the suggestion of Reviewer 2, the text of the paper is proof read by a native English speaker. We are grateful to Reviewer 2 for technical suggestions that are now implemented in the text. The most part of questions rose by this Reviewer were addressed in the ms text, while some were commented in this Response letter. New references were added. We trust that all points, including the important data analysis questions, the definition of emotional stress, rose by Reviewer 2, are addressed in the manuscript now and that this has substantially improved our work.

Abstract, Introduction:  See comments regarding editing.

Following the suggestion of Reviewer 2, the text of the paper is proof read by a native English speaker

1.The stress resulting from ultrasonic frequencies being considered 'emotional' stress needs further elaboration

Following this recommendation of Reviewer 2, we have elaborated a bit more on the stress resulting from ultrasonic frequencies being considered 'emotional' stress, in the ms text:

“The term ‘emotional stress’ or similar definitions, i.e., ‘psychological stress’, ‘mental stress’, are often used to label a state of stress that is resulting from adversities in human life, such as death of relatives, divorce, humiliation or defeat, loss of a social status, deterioration of financial status (Oliveira et al., 2022; Taylor et al., 2018).This form of stress is considered as a response to any unpredictable and mentally devastating experience that sharply increases a risk for major depression and many psychiatric disorders. While other types of stressors in their majority are associated with negative emotions as well, such as, for example, increased signs of anxiety and depressive-like behaviours after physical stress, pain, externally induced increases of inflammatory factors or glucocorticoid levels, related states of stress are not triggered by a perception and processing of adverse information and do not involve a cognitive element as a key feature. Importantly, emotional stress can provoke pro-inflammatory changes in key limbic structures (Serrats et al., 2017; Shields et al., 2016) and systemically in the blood (Murdaca et al., 2022)”.

Obviously, it is not fully possible to mimic the neurobiology of ‘emotional stress’ in animals (e.g., primates) and much less feasible in small laboratory animals. In light of these views, we have earlier proposed a model of ‘emotional stress’ using ultrasound exposure of unpredictably emerging ‘emotionally negative’ 21-day long radiation in male mice (Costa-Nunes et al., 2020; Gorlova et al., 2019; Pavlov et al., 2019). We showed that an exposure of mice to ultrasound that unpredictably alternates its frequencies between 20-25 kHz, which corresponds to their vocalization to a ‘negative emotional state’, and frequencies of 25-45 kHz that are associated with a ‘neutral’ emotional state (Kuraoka and Nakamura, 2010). The application of the ultrasound stimulation, using the parameters indicated above, was based on well-established ethological observations. Although the nature of species-specific information transmitted by rodents at the ultrasonic range is not entirely clear. It has been found that mice display sensitivity to the sounds of the defined ranges of frequencies. As for instance, mice emit the sounds in a range of 20-25 kHz in life-threatening conditions, such as social defeat, pain and maternal separation (Borta et al., 2006; Hahn and Lavooy, 2005; Portfors, 2007; Takahashi et al., 2010) and thus, exposure to artificially generated ultrasound of these frequencies may offer closer analogy to major human stressors causing depression, in a mouse (Gorlova et al., 2019; Morozova et al., 2016; Pavlov et al., 2019; Sambon et al., 2021). The ultrasound exposure in male mice was associated with upregulated corticosterone level (Pavlov et al., 2019), activated hippocampal microglia, upregulated IL-1β and IL-6 locally in the hippocampus and in systemic circulation (Pavlov et al., 2019), increased content of a marker of oxidative stress protein carbonyl in limbic structures (Gorlova et al., 2019). Notably, ultrasound-exposed male mice developed prominent depressive-like and anxiety-like behavioral changes (Costa-Nunes et al., 2020; Gorlova et al., 2023, 2019; Pavlov et al., 2019; Strekalova et al., 2018) that were counteracted by chronic dosing with classic antidepressant fluoxetine (Morozova et al., 2016)“.

2.Are there any specific behavior-associated noise that the frequencies were attempting to replicate, ex. a conspecific alerting to nearby predators? 

According to well-established ethological observations, the ultrasound emitted by mice in frequencies between 20-25 kHz corresponds to a vocalization in a “negative emotional state”, and frequencies of 25-45 kHz that are associated with a “neutral” emotional state (Kuraoka and Nakamura, 2010). As for instance, mice emit the sounds in a range of 20-25 kHz in life-threatening conditions, such as social defeat, pain and maternal separation (Borta et al., 2006; Hahn and Lavooy, 2005; Portfors, 2007; Takahashi et al., 2010). The signals of 50 kHz and higher were found to be generated by mice and rats during physiologically positive experiences and are considered as a manifestation of animals’ states that are regarded as parallels of “positive emotions”. In particular, ultrasounds at this range of frequencies are emitted during mother-pup interactions, mating and positive social interactions (Branchi et al., 1998; Costantini and D’amato, 2005; Holy and Guo, 2005; Okabe et al., 2010; Panksepp et al., 2007). The threshold for upper level frequencies used in our work was therefore set to be 45 Hz or lower. Interestingly, the emission and perception of the ultrasounds were shown to be influenced by such factors, as social structure / interactions and a territorial context and can implicate complex patterns of sound emissions (Costantini and D’amato, 2005).

  1. Use of the word 'emotion' needs to be used with many qualifiers when working with animals. 

We fully agree with this view of Reviewer 1 and now have addressed the definition of the word 'emotion' in a context of stress paradigm used in our work, more in a detail. We hope that extended explanations have helped to define the dimensions that are appropriate in a field of experimental work on rodents. 

4.While the previous study only used male mice, that is not enough justification to only use female offspring in this study, as the previous study did not examine males following in utero exposure.  As the present paper presents a different paradigm, both sexes should be used.  No direct comparison can be made between the current study and the previous, male-only, study.

We fully agree with Reviewer 1 that it would be ideal to examine male offspring as well. It is correct that the previous study did not examine males following in utero exposure. We plan to examine both sexes in our next study with the ultrasound exposure of pregnant dams in employed here paradigm, that unfortunately, could not be done here for technical reasons. We fully accept the criticism expressed by Reviewer 1 in this connection, as well as a remark that no direct comparison can be made between the current study and the previous, male-only, study. We are conscious that the argument that the sex factor introduces great complication in the interpretation of the data, first of all, of behavioural results, cannot fully justify the exclusion of male offspring from the study, and look forward to address the question outlined  in this comments soon.

  1. Methods:  See comments regarding editing.  Mice being housed individually during ultrasound stress part - does this mean only during the ultrasound stress application?  Does it mean throughout all days within the time frame of the ultrasound stress? 

Thank you for this question, all mice were housed individually throughout the whole duration of all manipulations. The rationale for single housing was to provide proper settings for a two-bottle sucrose preference test that was employed in stressed mice to study their sensitivity to reward, a key sign of depressive-like syndrome. Besides, the use of single housing in mice is beneficial for a reduction of inter-individual variability in the measures of emotionality, and in triggering social interactions. The latter was one of the main focus of this work, as we have anticipated autistic-like changes in behaviour of challenged offspring.

  1. Was anything done to compensate/take into account the stress that comes from being singly housed? 

We fully accept the point that since single housing alone can cause a distress in female mice, it should be compensated. For that propose, we housed all mice in enriched environment using fascial tissue and wooden objects. We also refrained from the use of ventilated cages so animals could be exposed to olfactory cues from each other. These methodological details are added to the Methods section.

  1. Were there any control, non-injected animals?  Injections, even of saline, give the animals considerable stress in and of themselves.  As this paper is intending to examine the result of emotional stress from very specific sources, care needs to be taken to reduce and control for stress from other sources. 

Thank you for raising this important question as for possible stress impact of an intraperitoneal injection per se in our study. Indeed our former works support the view of this Reviewer that intraperitoneal injections can be a major stress for mice (Costa-Nunes et al., 2015). Meanwhile, in the current experiment, in a sake of a reduction of animal use, and because we trust that the effects of two intraperitoneal administration are likely to be overwritten by the effects of LPS,we chose not to use saline-injected controls. Appropriate limits in the number of mice to be studied in one experimental are determined not only by ethical considerations, but also by limited feasibility of behavioral techniques when large cohorts of animals have to be used. For reliable accuracy with behavioural studies, it is critical to provide comparable physiological conditions to all experimental animals, such as light cycle and other settings. This alone can put considerable limits on the timing of the study, and thus, group sizes, as all mice have to be studied on the same day. Also, we trust that the effects of only two intraperitoneal injections of a vehicle in control group of animals are most likely to fade away over prolonged time of the experiment. All injections have been done with individual needles to minimize a discomfort of animals as much as possible. In general, we fully accept the criticism that care needs to be taken to reduce and control for stress from other sources while intending to examine the result of emotional stress.  

  1. Open field test - mice are prey animals; they prefer to be as sheltered as possible, meaning that the more anxious mouse would spend more time around the edge of the open field, while the less anxious mouse is more likely to spend time in the central area.  What parameters were scored on the elevated plus maze?

We fully agree with this comment, as for the reason that mice are prey animals that tend to display thigmotaxis in open areas, we have scored time spent in the central area of the open field as possible measure of anxiety-like behaviour. In the elevated plus maze, that are classical tests for anxiety-like behaviour in laboratory rodents, we have recorded the number of open arm entries and total time spent in the open arms. We have used these measures of anxiety like behaviour in female mice as they were previously established in our laboratories (Veniaminova et al., 2017).

9.Section 4.9 describes procedures for both mouse and rat (line 401).

This typo is fixed now, we thank Reviewer 2 for bringing this error to our attention.

  1. Results:  See comments regarding editing.  Figure 1 is not mentioned in the text.

Thank you for this remark; we made sure that Figure 1 representing the study design is mentioned appropriately in the ms text.

  1. The n's for groups are both small and substantially different. 

We appreciate the concern of Reviewer 2 that n’s for groups could be probably larger. Meanwhile, we trust that the number of mice used in the first study with adult mice is quite in a range of commonly applied practice (7-8) and similar to group sizes with previously reported ultrasound stress experiments with maleC57BL6  mice (Gorlova et al., 2019; Pavlov et al., 2019; Sambon et al., 2020). As for the experiments with pups born by ultrasound exposed dams, n was equal 10 in the former ms version, and as now we include all mice from the study in the analysis (please see p. 14, 16), n is equal 14-18 mice per group. We agree however that groups sizes in the study with LPS could be larger.  However, we had general limitation with the total number of animals that we could study behaviourally, since all mice were supposed to be investigated simultaneously. Because we have expected larger variability with the ultrasound exposure, more animals was used in this experiment than in the study with LPS challenge, whose effects we have extensively studied earlier (Couch et al., 2016; Sambon et al., 2020; Schapovalova et al., 2022; Strekalova et al., 2021; Trofimov et al., 2017). 

We admit that these limitations have led to a not perfectly optimal study design in terms of equality / similarity of group sizes but we feel that our choice in this respect was reasonable and justifiable.

  1. Where is the paradigm for the offspring?  Figure 1 suggests there were three sets - ultrasound exposure, pregnancy and ultrasound exposure undergoing one set of behavioral tests, pregnancy and ultrasound exposure with a different set of behavioral tests.  Description within methods does not align with this figure. 

Thank you for this raising the issue with not sufficient clarity in the representation of the study design on Figure 1. Now we have substantially revised a description of the study design in the last para of the Introduction section and in the Figure 1 legend (please see also p.14 of this Response letter). The paradigm for the offspring is indicated underneath of the horizontal line that is representing general flaw of each experimental run B and C. We feel that more consisted representation of the experimental scheme is of advantage for easier perception of the content by a potential reader. We also have checked the description of the methods to make sure that it is align with Figure 1. We hope that Reviewer 2 can find a representation of the study design sufficiently clear now.

  1. Additionally, why were there no immune challenges in group A?  What was the point of group A? 

As it is indicated in revised Figure 1 legend and the last para of the Introduction, we first aimed to study the effects of the ultrasound on adult females, particularly, with respect to the induction of systemic pro-inflammatory changes that, as we have hoped, could be comparable to the effects of LPS administration. Therefore, group A was designed to address this question, and the first experiment used the non-pregnant female mice that have been exposed to the ultrasound stress paradigm.  This was done since up to now, we have only studied male mice in the ultrasound stress model.

14.Neither the chronic stress nor the behavioral tests, which were different than either groups B or C, were conducted at the same time as the other groups? 

We apologize that the study design was not presented with sufficient clarity. Indeed, two studies that are mentioned in this question of Reviewer 2 were different only with respect to a battery of behavioural tests used in the offspring. In the first study (B) we have investigated the emotionality of pups that were born from stress/LPS- challenged dams, in the second study  (C) we addressed the acquisition of special memories by pups born in another cohort of stress/LPS- challenged dams. Because of lengthy test procedures and potentially fading curve of effects of challenges, we chose to study emotionality and cognitive functions in two separate runs. This explanation is now added to the ms text, Figure 1 legend:

“Next, we studied behavioral changes in the offspring of ultrasound-exposed dams starting at day P60. These effects were compared against the effects of double LPS administration on the offspring of pregnant dams at embryonic days E11.5 and E12.5, in two separate runs (B) using a battery of tests for emotionality: Sucrose preference test, open field and elevated plus maze and (C) the Barnes maze. Chronic ultrasound exposure started same day the plugs were checked, plug day was set as day E0.5.”.

We hope that revised description of the study design provides sufficient clarity now.

15.Was there any analysis on pregnancy loss/litter size changes due to stressful experiences during pregnancy?

We thank Reviewer 2 for raising up this very important and interesting question, that unfortunately, was not examined in the current study. We will address this question in our next experiment.

16."The top 10 females with the most pronounced deficits were selected for data analysis."  This statement requires justification - was there a scientific, statistically relevant reason for removing certain animals?  How many animals were removed?  What were the criteria for 'most pronounced deficits'.  Until or unless this is sufficiently explained, no results reported in this paper can be considered an accurate reflection of the experiments performed.

We fully agree with Reviewer 2 raising this critical point regarding seemingly artificial limiting the sample size post data acquisition. We apologize for a typo in the methodology description since top female animals with the most pronounced changes in behaviour were meant to be selected for expensive deep sequencing assay (this experiment is still in progress); this part of the sentence was cut of. The outcome from the ongoing molecular study (not presented here) was planned to be relayed to behavioural results that would be exactly match reported here cohort. Second, equal sizes of groups is an important factor of the accuracy of statistical analysis that is particularly important in the analysis of parameters with high variability, such as rodent behaviour, as well as for bioinformatics used for gene expression profiling (not presented here). Again, we are conscious about scientific rationale of critical importance of inclusion of all results in the analysis. Following a criticism expressed in this comment, we have included all originally investigated groups  in the behavioural analysis of this part of the experiment. Respective changes were made in Methods, Results, Figures and Discussion. Of note, the overall outcome from behavioural study of pups did not change substantially.

  1. Section 2.1 - why is decreased sucrose preference considered a behavioral deficit?  In this section, time spent in the center was reduced, which is listed as an anxiety-like behavior, which is not the same as is described in other sections.  Figure 2, which links to section 2.1, mentions depressive-like syndrome, which is different from the anxiety-like behavior referenced in section 2.1.

Following criticism of Reviewer 2 we have edited a subheading of the p.2.2 and nw replaced this subheading

“Chronic ultrasound exposure induces behavioral deficits in adult female mice”

with a new one:

“Chronic ultrasound exposure induces behavioral abnormalities in adult female mice”.

Of note, while we agree that it is not quite accurate to consider anxiety-like behavior as behavioral deficit, anhedonia is considered as a deficit in a sensitivity to a reward. Indeed, time spent in the centre of the open field test is interpreted here as a sign of an anxiety-like behaviour, now were made sure that it is discussed consistently in other sections.

In response to a criticism of this reviewer that Figure 2, which represents the results that are discussed in the section 2.1, mentions only depressive-like syndrome, we have edited the title of the Figure, a former title:

“Chronic ultrasound stress induces depressive-like syndrome in female mice.”

is replaced with a new title:

“Chronic ultrasound stress induces depressive-like syndrome and elevated anxiety-like behavior in female mice.”

  1. Section 2.2 - were the animals used for this section the same reduced number that was used for the behavioral tests?  Were all animals used?  Were a different subset used?

Thank you for this question, now we have added this missing methodological detail in the Methods section:

We used 7-8 mice that were randomly selected from the same cohorts of animals that was studied for behaviour, for a subsequent ELISA analysis.”

Indeed, a choice as for group sizes was based on the recommendation of the kit manufacturing protocols (https://www.quansysbio.com/products-and-services/multiplex-assays/) and on our former successful experience with ELISA analysis that was used under similar experimental conditions (de Munter et al., 2020; Pavlov et al., 2019).

19.Sections 2.3, 2.4 - See the aforementioned comments regarding using only the top 10 different females.  Why is it sometimes 'behavioral deficits' and sometimes 'cognitive deficits'?

This question echoes with a criticism expressed by Reviewer 2 in a previous comment that was addressed in full in this Response letter (see p.16) and in the paper. We thank this Reviewer for bringing to our attention a discrepancy in the terms used; now we made sure that the term ‘cognitive deficits’ is employed systematically throughout the paper. According to the literature, this term is generally used to indicate impaired (compromised) cognitive functions. As for other forms of behavioural changes, we used the term ‘behavioural abnormalities’ and similar expressions.

  20.Figure 5 - what was cut off from the top of Figure 5A?  This figure would be better represented with two sided error bars.

A cut off from the top of Figure 5A is time elapsed of 120 s in the Barnes maze. In response to a suggestion of Reviewer 2, this figure is shown with two sided error bars that hopefully helps to  represent the data better.

  1. Discussion:  Conclusions are overstated due to previously discussed lack of reasoning for artificially limiting the sample size post data acquisition.  See previous comments regarding editing.  Consistency is needed between mentioning 'depressive' and 'anxiety' with regards to behavioral changes.  Again, decreased time in center of open field is considered a sign of increased anxiety-like behavior, in contrast to the methods section.  Is this paper intended to model depression, anxiety, ASD, neurodevelopmental disorders as a whole, or something else entirely?
  2. "Some offspring are resilient to certain environmental challenges" - this is also seen in humans.  As a result, removing these animals from data analysis prevents any results from these experiments from being extrapolated to humans.

This comment echoes with a criticism expressed by Reviewer 2 in p. 16 and p.19. As it is mentioned above, the issue with removing (potentially) resilient animals from data analysis is addressed in this Response letter (see p.16) and in the paper. All animals are included in the behavioural analysis now.

  1. Several areas of the paper are lacking in grammar, most often with regards to lack of particles and articles.  An example is line 18, in the abstract.  'Major limitation' should be 'Amajor limitation'.  This lack of connecting words is seen throughout the paper, indicating a need for grammatical editing.  Editing is also needed for consistency of word choices; ex. dams vs. moms, behaviour vs. behavior.

In response to this criticism, we have used a help of a native English speaker for grammatical editing.

References

Borta, A., Wöhr, M., Schwarting, R.K.W., 2006. Rat ultrasonic vocalization in aversively motivated situations and the role of individual differences in anxiety-related behavior. Behav. Brain Res. 166, 271–280. https://doi.org/10.1016/J.BBR.2005.08.009

Branchi, I., Santucci, D., Vitale, A., Alleva, E., 1998. Ultrasonic vocalizations by infant laboratory mice: a preliminary spectrographic characterization under different conditions. Dev. Psychobiol. 33, 249–256. https://doi.org/10.1002/(sici)1098-2302(199811)33:3<249::aid-dev5>3.0.co;2-r

Costa-Nunes, J.P., Cline, B.H., Araújo-Correia, M., Valencą, A., Markova, N., Dolgov, O., Kubatiev, A., Yeritsyan, N., Steinbusch, H.W.M., Strekalova, T., 2015. Animal models of depression and drug delivery with food as an effective dosing method: Evidences from studies with celecoxib and dicholine succinate. Biomed Res. Int. 2015. https://doi.org/10.1155/2015/596126

Costa-Nunes, J.P., Gorlova, A., Pavlov, D., Cespuglio, R., Gorovaya, A., Proshin, A., Umriukhin, A., Ponomarev, E.D., Kalueff, A. V., Strekalova, T., Schroeter, C.A., 2020. Ultrasound stress compromises the correlates of emotional-like states and brain AMPAR expression in mice: effects of antioxidant and anti-inflammatory herbal treatment. Stress 23, 481–495. https://doi.org/10.1080/10253890.2019.1709435

Costantini F, D’amato FR, 2005. Ultrasonic vocalizations in mice and rats: Social contexts and functions | Request PDF. Acta Zool Sin 52, 619–633.

Couch, Y., Trofimov, A., Markova, N., Nikolenko, V., Steinbusch, H.W., Chekhonin, V., Schroeter, C., Lesch, K.P., Anthony, D.C., Strekalova, T., 2016. Low-dose lipopolysaccharide (LPS) inhibits aggressive and augments depressive behaviours in a chronic mild stress model in mice. J. Neuroinflammation 13. https://doi.org/10.1186/S12974-016-0572-0

de Munter, J.P.J.M., Shafarevich, I., Liundup, A., Pavlov, D., Wolters, E.C., Gorlova, A., Veniaminova, E., Umriukhin, A., Kalueff, A., Svistunov, A., Kramer, B.W., Lesch, K.P., Strekalova, T., 2020. Neuro-Cells therapy improves motor outcomes and suppresses inflammation during experimental syndrome of amyotrophic lateral sclerosis in mice. CNS Neurosci. Ther. 26, 504–517. https://doi.org/10.1111/CNS.13280

Gorlova, A., Pavlov, D., Zubkov, E., Zorkina, Y., Inozemtsev, A., Morozova, A., Chekhonin, V., 2019. Alteration of oxidative stress markers and behavior of rats in a novel model of depression. Acta Neurobiol. Exp. (Wars). 79, 232–237. https://doi.org/10.21307/ane-2019-021

Gorlova, A., Svirin, E., Pavlov, D., Cespuglio, R., Proshin, A., Schroeter, C.A., Lesch, K.P., Strekalova, T., 2023. Understanding the Role of Oxidative Stress, Neuroinflammation and Abnormal Myelination in Excessive Aggression Associated with Depression: Recent Input from Mechanistic Studies. Int. J. Mol. Sci. 24. https://doi.org/10.3390/IJMS24020915

Hahn, M.E., Lavooy, M.J., 2005. A review of the methods of studies on infant ultrasound production and maternal retrieval in small rodents. Behav. Genet. 35, 31–52. https://doi.org/10.1007/S10519-004-0854-7

Holy, T.E., Guo, Z., 2005. Ultrasonic songs of male mice. PLoS Biol. 3, 1–10. https://doi.org/10.1371/JOURNAL.PBIO.0030386

Kuraoka, K., Nakamura, K., 2010. Vocalization as a specific trigger of emotional responses. Handb. Behav. Neurosci. 19, 167–175. https://doi.org/10.1016/B978-0-12-374593-4.00017-6

Morozova, A., Zubkov, E., Strekalova, T., Kekelidze, Z., Storozeva, Z., Schroeter, C.A., Bazhenova, N., Lesch, K.P., Cline, B.H., Chekhonin, V., 2016. Ultrasound of alternating frequencies and variable emotional impact evokes depressive syndrome in mice and rats. Prog. Neuro-Psychopharmacology Biol. Psychiatry 68, 52–63. https://doi.org/10.1016/j.pnpbp.2016.03.003

Murdaca, G., Paladin, F., Casciaro, M., Vicario, C.M., Gangemi, S., Martino, G., 2022. Neuro-Inflammaging and Psychopathological Distress. Biomedicines 10. https://doi.org/10.3390/BIOMEDICINES10092133

Okabe, S., Nagasawa, M., Kihara, T., Kato, M., Harada, T., Koshida, N., Mogi, K., Kikusui, T., 2010. The effects of social experience and gonadal hormones on retrieving behavior of mice and their responses to pup ultrasonic vocalizations. Zoolog. Sci. 27, 790–795. https://doi.org/10.2108/ZSJ.27.790

Oliveira, J.M.D. de, Butini, L., Pauletto, P., Lehmkuhl, K.M., Stefani, C.M., Bolan, M., Guerra, E., Dick, B., De Luca Canto, G., Massignan, C., 2022. Mental health effects prevalence in children and adolescents during the COVID-19 pandemic: A systematic review. Worldviews evidence-based Nurs. 19, 130–137. https://doi.org/10.1111/WVN.12566

Panksepp, J.B., Jochman, K.A., Kim, J.U., Koy, J.K., Wilson, E.D., Chen, Q., Wilson, C.R., Lahvis, G.P., 2007. Affiliative behavior, ultrasonic communication and social reward are influenced by genetic variation in adolescent mice. PLoS One 2. https://doi.org/10.1371/JOURNAL.PONE.0000351

Pavlov, D., Bettendorff, L., Gorlova, A., Olkhovik, A., Kalueff, A. V., Ponomarev, E.D., Inozemtsev, A., Chekhonin, V., Lesсh, K.P., Anthony, D.C., Strekalova, T., 2019. Neuroinflammation and aberrant hippocampal plasticity in a mouse model of emotional stress evoked by exposure to ultrasound of alternating frequencies. Prog. Neuro-Psychopharmacology Biol. Psychiatry 90, 104–116. https://doi.org/10.1016/j.pnpbp.2018.11.014

Portfors, C. V., 2007. Types and functions of ultrasonic vocalizations in laboratory rats and mice. J. Am. Assoc. Lab. Anim. Sci. 46, 28–34.

Sambon, M., Gorlova, A., Demelenne, A., Alhama-Riba, J., Coumans, B., Lakaye, B., Wins, P., Fillet, M., Anthony, D.C., Strekalova, T., Bettendorff, L., 2020. Dibenzoylthiamine has powerful antioxidant and anti-inflammatory properties in cultured cells and in mouse models of stress and neurodegeneration. Biomedicines 8. https://doi.org/10.3390/BIOMEDICINES8090361

Sambon, M., Wins, P., Bettendorff, L., 2021. Neuroprotective Effects of Thiamine and Precursors with Higher Bioavailability: Focus on Benfotiamine and Dibenzoylthiamine. Int. J. Mol. Sci. 22. https://doi.org/10.3390/IJMS22115418

Schapovalova, O., Gorlova, A., de Munter, J., Sheveleva, E., Eropkin, M., Gorbunov, N., Sicker, M., Umriukhin, A., Lyubchyk, S., Lesch, K.-P., Strekalova, T., Schroeter, C.A., 2022. Immunomodulatory effects of new phytotherapy on human macrophages and TLR4- and TLR7/8-mediated viral-like inflammation in mice.

Serrats, J., Grigoleit, J.S., Alvarez-Salas, E., Sawchenko, P.E., 2017. Pro-inflammatory immune-to-brain signaling is involved in neuroendocrine responses to acute emotional stress. Brain. Behav. Immun. 62, 53–63. https://doi.org/10.1016/J.BBI.2017.02.003

Shields, G.S., Kuchenbecker, S.Y., Pressman, S.D., Sumida, K.D., Slavich, G.M., 2016. Better cognitive control of emotional information is associated with reduced pro-inflammatory cytokine reactivity to emotional stress. Stress 19, 63–68. https://doi.org/10.3109/10253890.2015.1121983

Strekalova, T., Bahzenova, N., Trofimov, A., Schmitt-Böhrer, A.G., Markova, N., Grigoriev, V., Zamoyski, V., Serkova, T., Redkozubova, O., Vinogradova, D., Umriukhin, A., Fisenko, V., Lillesaar, C., Shevtsova, E., Sokolov, V., Aksinenko, A., Lesch, K.P., Bachurin, S., 2018. Pro-neurogenic, memory-enhancing and anti-stress effects of DF302, a novel fluorine gamma-carboline derivative with multi-target mechanism of action. Mol. Neurobiol. 55, 335–349. https://doi.org/10.1007/s12035-017-0745-6

Strekalova, T., Svirin, E., Waider, J., Gorlova, A., Cespuglio, R., Kalueff, A., Pomytkin, I., Schmitt-Boehrer, A.G., Lesch, K.P., Anthony, D.C., 2021. Altered behaviour, dopamine and norepinephrine regulation in stressed mice heterozygous in TPH2 gene. Prog. Neuro-Psychopharmacology Biol. Psychiatry 108. https://doi.org/10.1016/j.pnpbp.2020.110155

Takahashi, N., Kashino, M., Hironaka, N., 2010. Structure of rat ultrasonic vocalizations and its relevance to behavior. PLoS One 5. https://doi.org/10.1371/JOURNAL.PONE.0014115

Taylor, H.O., Taylor, R.J., Nguyen, A.W., Chatters, L., 2018. Social Isolation, Depression, and Psychological Distress Among Older Adults. J. Aging Health 30, 229–246. https://doi.org/10.1177/0898264316673511

Trofimov, A., Strekalova, T., Mortimer, N., Zubareva, O., Schwarz, A., Svirin, E., Umriukhin, A., Svistunov, A., Lesch, K.P., Klimenko, V., 2017. Postnatal LPS Challenge Impacts Escape Learning and Expression of Plasticity Factors Mmp9 and Timp1 in Rats: Effects of Repeated Training. Neurotox. Res. 32, 175–186. https://doi.org/10.1007/S12640-017-9720-2

Veniaminova, E., Cespuglio, R., Cheung, C.W., Umriukhin, A., Markova, N., Shevtsova, E., Lesch, K.P., Anthony, D.C., Strekalova, T., 2017. Autism-Like Behaviours and Memory Deficits Result from a Western Diet in Mice. Neural Plast. 2017. https://doi.org/10.1155/2017/9498247

Round 2

Reviewer 1 Report

I thank the authors for considering my questions and suggestions.

The manuscript improved.

In any case, I suggest authors review statistical concepts for future studies, especially related to the calculation of sample size and the magnitude of the effect. With a very low n (n=5) the statistical power is low and it can be inferred that there are no differences, but in reality the low statistical power does not allow this statement to be made. Therefore, it is important to be cautious with the inferences, and for that it is important to know the statistical power. In the same way with the size of the effect, an element is what has to do with the difference between groups, which is evaluated with the p-value, as the degree of error of rejecting the null hypothesis. But the size of the effect goes in another direction, and precisely shows the biological value of the possible differences between groups. I recommend reviewing these concepts.

Author Response

Following the recommendation of Reviewer 1 we have subjected the ms text to an additional revision of the text that is done by a native English speaker. We trust that we have done at most to fulfill all the requests and recommendations of Reviewers while revising the paper.

Reviewer 2 Report

While the points raised in the previous review have begun to be addressed, further clarification is needed.  It is recommended that the draft undergo another round of review.

While there have been improvements regarding the editing, word choices, and grammar, another review of these is needed.  Ex. Line 278 Neuroninflammation vs. Neuroinflammation.

Author Response

Following the recommendations of Reviewers 1 and 2 we have subjected the ms text to an additional revision of the text that is done by a native English speaker. We trust that we have done at most to fulfill all the requests and recommendations of Reviewers while revising the paper.
